# Antiviral Mechanisms of *N*-Phenyl Benzamides on Coxsackie Virus A9

**DOI:** 10.3390/pharmaceutics15031028

**Published:** 2023-03-22

**Authors:** Mira Laajala, Kerttu Kalander, Sara Consalvi, Olivier Sheik Amamuddy, Özlem Tastan Bishop, Mariangela Biava, Giovanna Poce, Varpu Marjomäki

**Affiliations:** 1Department of Biological and Environmental Science/Nanoscience Center, University of Jyväskylä, 40014 Jyväskylä, Finland; mira.a.laajala@jyu.fi (M.L.); kerttu.kalander@helsinki.fi (K.K.); 2Department of Chemistry and Technologies of Drug, Sapienza University of Rome, Piazzale A. Moro 5, 00185 Rome, Italy; sara.consalvi@uniroma1.it (S.C.); mariangela.biava@uniroma1.it (M.B.); giovanna.poce@uniroma1.it (G.P.); 3Research Unit in Bioinformatics (RUBi), Department of Biochemistry and Microbiology, Rhodes University, Makhanda 6140, South Africa; oliserand@gmail.com (O.S.A.); ozlem.tastanbishop@gmail.com (Ö.T.B.)

**Keywords:** enterovirus, antiviral, capsid binder, *N*-phenyl benzamide

## Abstract

Enteroviruses are one of the most abundant groups of viruses infecting humans, and yet there are no approved antivirals against them. To find effective antiviral compounds against enterovirus B group viruses, an in-house chemical library was screened. The most effective compounds against Coxsackieviruses B3 (CVB3) and A9 (CVA9) were CL212 and CL213, two *N*-phenyl benzamides. Both compounds were more effective against CVA9 and CL213 gave a better EC_50_ value of 1 µM with high a specificity index of 140. Both drugs were most effective when incubated directly with viruses suggesting that they mainly bound to the virions. A real-time uncoating assay showed that the compounds stabilized the virions and radioactive sucrose gradient as well as TEM confirmed that the viruses stayed intact. A docking assay, taking into account larger areas around the 2-and 3-fold axes of CVA9 and CVB3, suggested that the hydrophobic pocket gives the strongest binding to CVA9 but revealed another binding site around the 3-fold axis which could contribute to the binding of the compounds. Together, our data support a direct antiviral mechanism against the virus capsid and suggest that the compounds bind to the hydrophobic pocket and 3-fold axis area resulting in the stabilization of the virion.

## 1. Introduction

Enteroviruses belong to the family of picornaviruses, which includes 15 species of viruses in total (enteroviruses A–L, rhinoviruses A–C), but only 7 species can infect humans. They are around 30 nm in diameter and have a single-stranded RNA genome of approximately 7500 base pairs. The symptoms caused by enterovirus infections are usually mild, but they have also been associated with more serious diseases like type 1 diabetes, heart attack and acute stroke [1,2,3].

Coxsackievirus A9 (CVA9) and Coxsackievirus B3 (CVB3) serotypes both belong to the enterovirus B species. Although both viruses often cause mild diseases, CVA9 has also been shown to cause meningitis as well as hepatitis [4,5], while CVB3 is more associated with myocarditis, pancreatitis as well as meningitis [6].

Enteroviruses enter through the fecal-oral route and their primary infection site is the gastrointestinal epithelium. In addition, rhinoviruses and enterovirus D68 can spread through respiratory secretions and replicate in the respiratory tract [7]. After the initial replication, the viruses can spread to secondary tissues such as the skin, heart, or pancreas [8].

Antivirals are needed as an alternative way to combat enterovirus infections besides vaccines. The most promising results in the search for effective antivirals have thus far been obtained only with pleconaril and vapendavir, both are capsid-binding inhibitors, which faced severe drawbacks such as interference with metabolic pathways of other drugs and drug resistance [9]. Therefore, further efforts are needed to develop novel antivirals. In this context, we have screened an in-house chemical library of 200 compounds of diverse chemical scaffolds, previously synthesized in our lab, against CVA9 and CVB3. The structural classes represented have a history of antimicrobial, antimycobacterial, and anti-inflammatory activities and fall into different categories:  pyrroles, pyrazoles, imidazoles, *N*-phenyl benzamides, toluidines, phenyl cinnamamides, methanimines, triazoles, F-anilines, phenyl hydrazides, and furane (details and chemical structures of the molecules are presented in Appendix A). Among the screened compounds two *N*-phenyl benzamides showed outstanding activity against CVA9. In this paper, we describe the antiviral screening and, for those molecules showing potential, the study of their inhibition mechanism and efficacy.

## 2. Materials and Methods

### 2.1. Chemistry

All the tested compounds belong to an in-house library of molecules previously prepared. The chemical identity of the compounds was assessed by re-running NMR experiments. The purity of all compounds, checked by reversed-phase High-Performance Liquid Chromatography (HPLC), was always higher than 95%. Synthetic procedures for CL212 and CL213 are reported in the Appendix A.

### 2.2. Cells, Viruses, and Molecules

A549 lung carcinoma cells (ATCC) were cultured in Dulbecco’s Modified Eagle Medium (DMEM, Thermo Fisher, Waltham, MA, USA) supplemented with 10% Fetal bovine serum (FBS, Gibco, Life Technologies, Paisley, UK), antibiotics (1% penicillin and streptomycin, Gibco, Life Technologies, Paisley, UK), and 1% Glutamax (Gibco, Life Technologies, Paisley, UK).

For experiments, the cells were plated on 96-well plates (Sarstedt, Nümbrecht, Germany) with 12,000 cells per well. The edge wells of the plates were not used for samples to avoid edge effects. The viruses used in the experiments were CVA9 (Griggs strain, ATCC) and CVB3 (Nancy strain, ATCC).

The compounds were dissolved into DMSO to create 20 mM stocks and stored at −80 °C. Altogether there were 200 compounds, which were labeled as CL plus a given number which ranged from 1 to 220.

### 2.3. Concentration Series, EC_50_ and CC_50_

The compounds were tested against both CVA9 and CVB3 at six concentrations with three repeats of each (1 µM, 10 µM, 20 µM, 50 µM, 75 µM and 100 µM). The series contained three cell plates, one of which was infected with CVA9, one with CVB3, and one control plate which was left without virus infection. Plates had pleconaril controls (10 µM), and cell controls, and due to the molecules having been dissolved into DMSO, controls of matching DMSO dilutions to the molecules were made to see whether DMSO affected the cell viability. Tested compounds were diluted in DMEM supplemented with 1% FBS and incubated with viruses (1.6 × 10^7^ PFU/mL, MOI 80) at 37 °C, 5% CO_2_ for 1 h. The mixtures were then transferred onto cells and incubated overnight (37 °C, 5% CO_2_).

The infection was studied using a cytopathic effect (CPE) assay. After washing with PBS, the cells were stained with crystal violet-containing dye (0.03% crystal violet, 2% ethanol, and 3% formalin in H_2_O). Next, the cells were washed twice with sterile H_2_O to remove any excess dye, and finally, lysis buffer (47% ethanol, 12.5 mM HCl, and 19 mM sodium citrate) was added to homogenize the solution. Absorbance was measured at 570 nm with a multi-plate reader (VictorTM X4, PerkinElmer, Waltham, MA, USA).

The data were normalized against the absorbance of the cell control which was set to 100%.

EC_50_ and CC_50_ value determination against CVA9 (MOI 80) was carried out with Graph pad prism (version 6.07, Dotmatics, San Diego, CA, USA) using non-linear regression analysis with a four-parameter model.

### 2.4. Time-of-Addition Assay

For time-of-addition assays, compounds were (1) only present during a one-hour pre-incubation with the virus, (2) present the whole time, and (3) present from one-hour post-infection (p.i.) onwards. Compounds were diluted into DMEM (supplemented with 1% FBS) to a concentration of 100 µM (CL212) and 10 µM (CL213). In treatment (1), compounds were first incubated with the virus (CVA9 and CVB3, 1.6 × 10^7^ PFU/mL) for one hour at 37 °C, 5% CO_2_. Next, the virus/molecule mixture was added onto cells and incubated on ice for 1 h. After washing the excess virus away, fresh DMEM supplemented with 1% FBS and 1% glutamax was added to the cells and the infection continued overnight. Treatment (2) was otherwise the same as (1) except compounds were present the whole infection time as the fresh medium also contained the compounds. In treatment (3) compounds were added to the cells only after the infection had gone on for one hour at 37 °C, 5% CO_2_. Cell viability was assessed using a CPE assay (as described above). The data was normalized against the cell control after which the viabilities from each sample were compared to the lowest virus control viability which was set to zero.

For qPCR and confocal microscopy studies, the compounds (10 µM or 100 µM) were incubated with the viruses (1.6 × 10^7^ PFU/mL) for 1 h before adding the mixture to the cells for 1 h of ice-binding. After the excess virus was washed away, the infection was allowed to proceed at +37 °C for 5.5 h. In post-infection samples, the compounds were added 2 h after the infection was started at +37 °C. The RNA isolation and qPCR were carried out as described before [10]. Confocal microscopy samples were fixed with 4% PFA and immunolabeling using an antibody against VP1 capsid protein (a kind gift from Dr. Merja Roivainen) was carried out as described earlier [10]. Imaging was carried out using Olympus FV1000-IX81 confocal microscope as described earlier [10].

### 2.5. Real-Time Fluorescence Uncoating Assay

The real-time uncoating assay, using CVA9 and CVB3 was performed as previously described [11]. Compounds (100 µM in 2 mM MgCl_2_/PBS) and virus (0.5 µg) were added into each sample and incubated for one hour at 37 °C, 5% CO_2_. In an additional control assay, the viruses were incubated in an opening buffer (20 mM NaCl, 6 mM KH_2_PO_4_, 12 mM K_2_HPO_4_, and 0.01% faf-BSA) or in a storage buffer (2 mM MgCl_2_/PBS) as described above. 10× RNA/DNA intercalating dye, SYBR green II (InvitrogenTM) was added into each sample, and RNAase (10 mg/mL, Thermofisher Scientific, EN0531) to half of the samples as a control.

Fluorescence was measured with a multi-plate reader (VictorTM X4, PerkinElmer, Waltham, MA, USA) five times with 15 min intervals. The background fluorescence of the buffer was subtracted from the virus samples. In addition, the fluorescence of the samples which contained RNAase was deducted from the corresponding sample without the RNAase to end up with fluorescence values of only the virus-released RNA. Finally, the fluorescence values were normalized against the 15 min virus control which was set to 1.

### 2.6. Radioactive Gradient Assay

The effect of CL212 and CL213 on the virus capsid was studied with 35S-methionine/cysteine-labeled CVA9 as described before [12]. Metabolically labeled CVA9 (1.9 × 10^7^ PFU/mL with 75 CPM) was incubated with 100 µM CL212 or CL213 in 2 mM MgCl_2_/PBS for one hour at 37 °C, 5% CO_2_. Next, the samples were transferred on top of a linear sucrose gradient (20–5%, in R-buffer with 10 mM Tris-HCl, 0.2 M NaCl, 50 mM MgCl_2_, 10% glycerol in H_2_O). After centrifugation for two hours at 35,000 rpm (Optima LE-80K Ultracentrifuge, Beckman, USA, with rotor SW41, Beckman, Ireland), 500 µL fractions were collected from the top and mixed with a scintillation cocktail (ULTIMA GOLD^TM^ MV, PerkinElmer, Waltham, MA, USA). Finally, the radioactivity of each fraction was measured with Liquid Scintillation Analyzer (Tri-Carb^®^ 2910 TR, PerkinElmer, Downers Grove, IL, USA).

### 2.7. Transmission Electron Microscope (TEM) Imaging

CL212 and CL213 (100 µM) were incubated with CVA9 (1 µg) in 2 mM MgCl_2_/PBS for one hour at 37 °C, 5% CO_2_, and then transferred onto glow discharged TEM grids and dyed with phosphotungstic acid (1% solution in H_2_O) as described earlier [12]. The samples were imaged with JEOL JEM-1400 (80 kV, quartet holder).

### 2.8. Docking Imaging

3D structures from CL212 and CL213 were drawn from their 2D representations before being energy-minimized (with default settings) using Avogadro (version 2, Open Chemistry, New York, NY, USA) [13]. Capsid portions containing the 2-fold and 3-fold axes of symmetry in both the CVA9 Griggs strain (PDB ID: 1D4M) and CVB3 Nancy strain (PDB ID: 6GZV) were prepared such that the individual docking grids were centered at the axes of symmetry, spanning the outer surfaces while being partly buried within the capsid. The capsid portions were preprocessed using the “prepare_receptor4.py” script from MGLTools while the ligands were preprocessed using the “prepare_ligand4.py” script, whereby Gasteiger partial charges were assigned to all the atoms, and non-polar hydrogen atoms were merged [14]. QuickVina-W was then used with an exhaustiveness of 1000 and a grid box interval size of 0.375 Å, to blindly dock the compounds against the capsid portions [15]. Docking runs were performed at the Center for High Performance Computing, as detailed previously [16]. All docked poses were retained and their binding energies were recorded and plotted. The poses were also visualized using the PyMOL Autodock/Vina plugin [17], while the PyVOL plugin (version 1.6.8, Schlessinger Lab, New York, NY, USA) was used to highlight the hydrophobic pockets. Further details are provided in our previous work [16].

### 2.9. Statistical Analysis

Samples were compared pairwise using a *t*-test. Before the *t*-test, arcsin √ transformation of the original variable was applied for ratio or percentage data to convert the binomial distribution of the data to follow a normal distribution. Two-tailed *p*-values were used for the assessment of the significance between virus controls and compound samples at the time of addition assay. In the real-time fluorescence uncoating assay, one-tailed *p*-values from comparisons between virus control and compound samples were assessed due to there already being the assumption that the compounds would decrease uncoating.

## 3. Results

### 3.1. Potential Antiviral Identification

In order to find effective compounds directly acting on enteroviruses, initial screens with selected 200 compounds were performed using pretreatment of the enteroviruses CVA9 with 10 µM compound concentrations (Appendix A). The two most potent compounds, CL212 and CL213 (Figure 1) were selected for further testing. For these molecules, a wider concentration series against CVA9 and CVB3 was performed to further assess their effectivity and cytotoxicity (Figure 1). Remarkably, both compounds showed no apparent cytotoxicity even at the highest tested concentration of 100 µM. Both drugs worked efficiently against CVA9 infection reaching high cell viability (70 and 80% for CL212 and CL213, respectively) already at 10 µM concentration. In contrast, the Nancy strain of CVB3 was not efficiently inhibited as the cell viability reached only approximately 30%.

For CL212 and CL213, half maximal effective concentration (EC_50_), 50% Cytotoxic Concentration (CC_50_), and selectivity indexes (SI) were determined against CVA9 (1.6 × 10^7^ PFU/mL) (Figure 2, Table 1). As for CL212, fully cytotoxic concentrations were not reached, SI was calculated using 1 mM as the cytotoxic concentration. Both molecules had high SI values of over a hundred.

### 3.2. Initial Structure Activity Relationship (SAR) Studies

To build up initial SARs, an analysis of the chemical structures of the in-house library in searching for chemical analogs of both CL212 and CL213 was carried out. The in-house chemical library contains 13 *N*-phenyl benzamides and 4 amine analogs of CL212 and CL213 (Figure 3), through the analysis of which, preliminary SARs were built up. Briefly, the phenyl benzamide moiety seems essential for the activity since amines CL30, CL34, CL35, and CL215 showed the same cell viability as the virus control (Appendix A). The same is true for CL3, CL4, CL10, CL9, CL6, CL32, CL214, CL30, CL34, CL35, and CL215, which highlights that the replacement of the *N*-methyl piperazinyl moiety with the morpholinyl, tiomorpholinyl, or imidazolyl ones causes a loss of activity. Moreover, the *N*-methyl piperazinyl ring seems essential at *para* position.

### 3.3. CL212 and CL213 Prevent the Infection Already after Pre-Incubation

To find out at which point of the virus life cycle the inhibitors show antiviral effect, time-of-addition assays were performed with both CVA9 and CVB3. In the assay, it was tested if the molecules are effective directly on the virions, effective when added after virus entry to the cells, or when being present all the time (see Materials and Methods for details). The CPE-based assay demonstrated that CL212 (100 µM) inhibited CVA9 and CVB3 infections most efficiently when it was present the whole infection time, while with CL213 (10 µM), the best efficacy was obtained with the pre-incubation (Figure 4A). qPCR-based assay measuring of the infection during the first 5.5 h further confirmed that for both viruses and compounds, the pre-treatment showed the best protection against virus infection (Figure 4B). Confocal microscopy further showed that the pre-treatment of CVA9 with CL213 was the most efficient as the infection percentage was only 4.2% (*n* = 259) while in control infection and post-treatment, 25% (*n* = 259) and 23% (*n* = 221) of cells were infected, respectively. Furthermore, confocal microscopy showed that pre-treatment allowed entry of CVA9, but the virus was blocked in endosomes (Figure 4C).

Altogether, the results showed that both compounds were able to inactivate the tested enteroviruses already after pre-incubation, but molecules had a contributing effect later in the infection as well.

### 3.4. CL212 and CL213 Stabilize the Virus Capsid

#### 3.4.1. Uncoating

To reveal if CL212 and CL213 affect the uncoating of the enteroviruses, the in-house developed real-time uncoating assay was performed. The uncoating of the virus would be seen in the assay as an increased fluorescence as the virus genome is released and, on the contrary, low fluorescence would signify the virus staying intact [18]. As a control, we performed an uncoating assay of both viruses in a stabilizing storage buffer or in an opening buffer which has earlier been shown to enhance the uncoating of CVA9 [18] and which here increased the fluorescence 8-times or 2-times higher with CVA9 and CVB3, respectively (Figure 5A).

Both molecules incubated with CVA9 resulted in lower fluorescence than without the molecule throughout the whole 60 min measured at 15-min intervals (Figure 5B). These results thus indicate that both compounds kept CVA9 intact and prevented even the minor expansion of the virions seen for the control virus. Similarly, CVB3 treated with CL213 resulted in lower fluorescence values than control values, suggesting that CL213 stabilized also CVB3. Although CL212 did not similarly lower the fluorescence when added to CVB3, it did not, however, cause further opening of the virus as the fluorescence was not higher than the control values. In conclusion, neither of the molecules induced the opening of the virus but rather stabilized the capsid and prevented RNA release.

#### 3.4.2. Virus Stays Intact Detected by Sucrose Gradient Fractionation

As both CL212 and CL213 showed great efficacy on CVA9, we next used radioactive gradient separation which demonstrates the status of the virus in more detail: the intact/full (160 S), partially opened (135 S), and RNA-free, uncoated virus capsid (80 S). After incubating the radioactive CVA9 with the molecules for 1 h at 37 °C, the samples were subjected to a 5–20% sucrose gradient and fractionated from the top (Figure 6). As for the control, untreated virus showed the highest peak around the fractions 16–18 as expected, and no apparent peak around 8–10 fractions, suggesting no empty viruses in the control preparation. After treatment with both molecules, the peak of CVA9 was detected amongst the same fractions as with the control and no virions around 8–10 fractions, suggesting that the virus remained intact during the treatments without any signs of expansion or opening.

#### 3.4.3. Virus Stays Intact Detected by Electron Microscopy

In order to get more direct information on the CVA9 virions after treatments with CL212 and CL213, we performed negative staining in TEM (Figure 7). In all samples, the majority of viruses showed a white appearance against dark background with no signs of expansion or opening [18]. In addition, virions were evenly spread on the grid and neither of the compounds caused aggregation or clustering of the viruses.

### 3.5. CL212 and CL213 Bind to the Hydrophobic Pocket and 3-Fold Axis Pore

To further investigate capsid localizations of CL212 and CL213, both the 2-fold and 3-fold axes of symmetry in CVA9 and CVB3 were targeted by in-silico docking simulations (Figure 8 and Figure 9). While most of the CL212 and CL213 poses occupied the hydrophobic pocket found within the 2-fold axis of the symmetry grid box, they favored the larger hydrophobic pocket of CVA9 more, over the collapsed pocket of the CVB3 Nancy strain. CL212 poses mainly occupied the hydrophobic pocket: 55% (22/40) in CVA9 and 82.5% (33/40) in CVB3.

The 3-fold axis pore accommodated some CL212 poses: 35% (14/40) in CVA9 but less with CVB3, 7.5% (3/40). CL213 poses were also mostly distributed between the hydrophobic pocket (30%, 6/20) and 3-fold axis pore (35%, 7/20) in the case of CVA9, but in contrast, all the CL213 poses were located in the hydrophobic pocket of CVB3 (39/39).

Lower binding energies of both compounds (Figure 10) within the hydrophobic pocket of CVA9 indicate their higher affinity for this pocket compared to that of CVB3. Additionally, the generally lower binding energies of CL213 compared to CL212, suggest that CL213 binds more strongly to the capsid.

## 4. Discussion

Although much effort has been put into anti-enteroviral research over the years, none of the studied compounds have passed clinical trials [19]. As enteroviruses are still one of the most abundant viruses infecting humans, more antiviral strategies and new molecules need to be explored. In this study, we screened an in-house library of 200 synthetic molecules and found two potential compounds, CL212 and CL213 that prevented the infection of CVA9 with low micromolar concentrations. In addition, the molecules showed very low cytotoxicity against the studied A549 cells.

To get an idea of the antiviral mechanism, a time of additional study was conducted. Interestingly, both CL212 and CL213 showed efficacy against CVA9 either when incubated before infection yet CL212 also had a small effect when added 1 h p.i. In addition, confocal microscopy showed that the molecules did not hinder receptor binding and internalization but allowed entry and blocked infection in endosomes. Further in vitro studies also suggested that CL212 and CL213 acted as capsid binders as the uncoating of the virus was prevented in the presence of the compounds and according to TEM imaging, the particles were intact. We have shown earlier that the uncoating of CVA9 starts after 30 min but takes place mostly between 1–2 h p.i. [20]. Thus, it is likely that the addition of stabilizing molecules even after 1 h p.i. was able to prevent the uncoating of the virus and thus infection. All in all, the studies thus showed that the antiviral effect could be achieved with a pre-incubation only, indicating that the molecules acted as capsid binders. However, other contributing effects later in infection cannot be ruled out.

Docking studies suggest that the majority of the CL212 and CL213 docking poses localize into the hydrophobic pocket of both viruses. For many years the best-identified area in the enterovirus capsid occupying antiviral molecules has been the hydrophobic pocket [21,22,23,24,25,26]. However, recently also another pocket was found by groups of Sarah Butcher and Johan Neyts to bind antiviral molecules against enteroviruses and rhinoviruses [27]. In addition, our earlier in-silico docking studies have suggested that an antiviral polyphenol resveratrol, could occupy the 3-fold axis pore of CVA9 [16]. It is not a surprise that the most potential binding site for both CL212 and CL213 was the hydrophobic pocket. Interestingly, the other experimentally proven binding site, namely the Butcher-Neyts pocket, did not occupy any of the poses, but instead, the 3-fold axis pore seemed to be a potential binding site with low binding energies. Although the 3-fold axis pore has not been proven experimentally to be a binding site, in the light of our earlier results with resveratrol, it cannot be ignored and should be studied further.

Although the most potential binding site in both viruses was the hydrophobic pocket, both molecules fitted better in the pocket of CVA9. In addition to the visual inspection, also the higher binding energies suggested that the molecules did not fit well into the pocket of CVB3. As the Nancy strain of CVB3 has been shown to be resistant to pleconaril, another capsid binder, due to an amino acid substitution in the VP1 inside the pocket [21,28,29], poorer binding and lower antiviral effect with CL212 and CL213 was not unexpected. Therefore, the results with the Nancy strain of CVB3 act as indirect proof of the hydrophobic pocket being the target of the molecules.

When new antiviral molecules are developed, important initial parameters that are studied in vitro include the EC_50_ and CC_50_ values of the molecules as well as the ratio of the efficacy versus cytotoxicity, namely SI. Although a rather high dose of the molecules is suggested by our study (1–2 µM), low cytotoxicity makes these, especially CL213 a great molecule to develop further. Physicochemical properties, pharmacokinetic properties, and the drug-likeness of both CL212 and CL213, calculated using SwissADME and given in the Appendix A, show that both compounds are endowed with good drug-likeness properties. Both compounds show zero violation of the Lipinski, Ghose, Veber, Egan, and Muegge rules, moreover, compounds can be promising agents that can very easily be absorbed by the gastrointestinal tract.

In conclusion, we showed that two *N*-phenyl benzamides, CL212 and CL213, efficiently inhibited the infection of CVA9 with low micromolar concentration. We further demonstrated that the compounds acted as capsid binders with suggested localization into the hydrophobic pocket and 3-fold axis area.

## Figures and Tables

**Figure 1 pharmaceutics-15-01028-f001:**
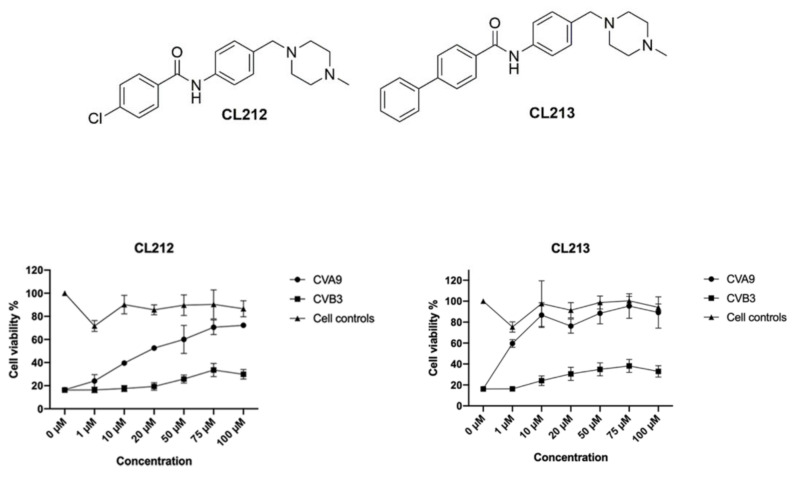
CL212 and CL213 rescue A549 cells from CVA9 infection. The viruses (1.6 × 10^7^ PFU/mL) were pre-treated with the molecules (CL212 or CL213) for 1 h at +37 °C and the next virus/molecule mixture was added to cells (MOI 80). Also, cytotoxicity controls were included with no virus (cell controls). Cell viability was determined after 24 h using CPE assay (see details in materials and methods). Viabilities are presented as mean ± SEM from three repeats.

**Figure 2 pharmaceutics-15-01028-f002:**
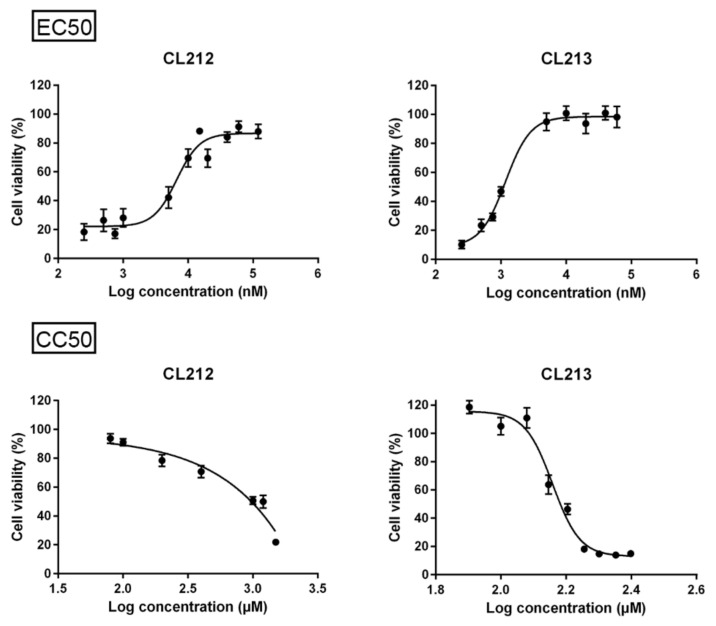
Dose-response curves of CL212 and CL213 to study their efficacy during CVA9 infection (upper panel) or cytotoxicity in A549 cells (lower panel). CVA9 (1.6 × 10^7^ PFU/mL) was incubated with the molecules for 1 h at +37 °C after which the virus-molecule mixture was added to cells (MOI 80). Cell viability was determined after 24 h using CPE assay (see materials and methods for details). The results are from two (EC50) or three (CC50) separate experiments with three technical replicates. The data is presented as mean ± SEM.

**Figure 3 pharmaceutics-15-01028-f003:**
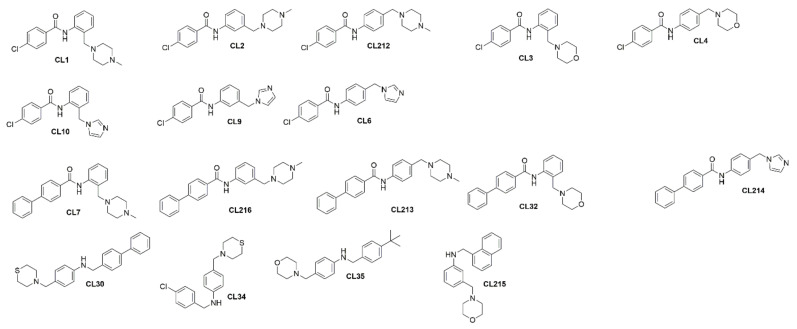
Chemical structures of CL212 and CL213 analogues.

**Figure 4 pharmaceutics-15-01028-f004:**
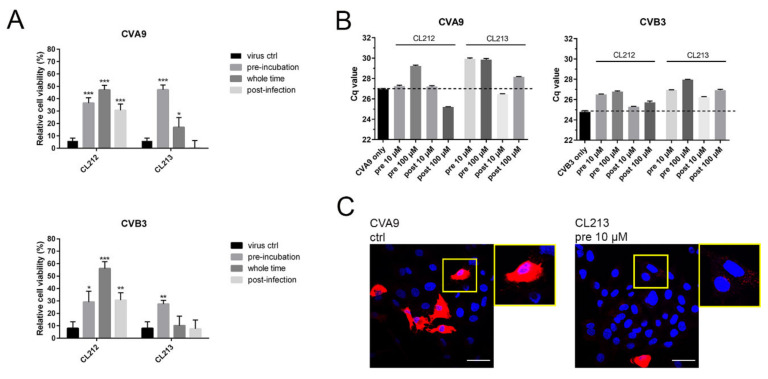
CL212 and CL213 prevent CVA9 infection already after pre-incubation but have less efficacy against the Nancy strain of CVB3. (**A**) Relative cell viabilities of cells infected with CVA9 or CVB3 (1.6 × 10^7^ PFU/mL) for 24 or 48 h after three different treatments with molecule CL212 (100 µM), or CL213 (10 µM against CVA9, 100 µM against CVB3). The molecules were either incubated with the virus for 1 h before infection, were added to cells 1 h post-infection or were present the whole pre-incubation and infection period. Virus control samples were with no molecule treatment. CVA9 graphs show viabilities from four (CL212) or three (CL213) separate experiments each containing three technical replicates. CVB3 graphs show viabilities from two separate experiments each containing three technical replicates. The data is normalized against an uninfected cell control and presented as mean ± SEM. From comparison to virus control samples significant p-values are marked with an asterisk above the respective sample columns: *p* < 0.05 *, *p* < 0.01 **, and *p* > 0.001 ***. (**B**) A549 cells were infected with CVA9 or CVB3 for 5.5 h and the amount of viral RNA was detected using qPCR. CL212 and CL213 were either pre-incubated with the viruses for 1 h before infection or were added 2 h p.i. (**C**) A549 cells were infected for 5.5 h with CVA9 with or without pre-incubating the virus with CL213 for 1 h. The infection was detected by immunolabeling the viral capsid protein VP1 (red). The cell nuclei were visualized using DAPI staining (blue). Scale bars 40 µm.

**Figure 5 pharmaceutics-15-01028-f005:**
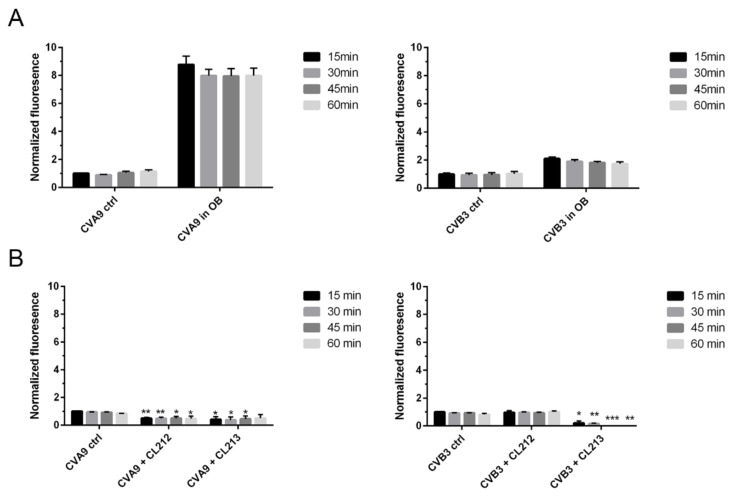
The effect of CL212 and CL213 on virus uncoating. CVA9 or CVB3 (0.5 µg) were incubated in stabilizing 2 mM MgCl_2_/PBS buffer (ctrl) or in an opening buffer OB (**A**), or in 2 mM MgCl_2_/PBS buffer with or without 100 µM molecules (**B**) for 1 h at +37 °C. After the incubation, Sybr Green II was added and fluorescence measurements were taken at 15-min intervals for one hour. The fluorescence is normalized against the 15 min ctrl virus and the data are presented as mean ± SEM from two repeats. From comparison to control samples significant *p*-values are marked with an asterisk above the respective sample columns: *p* < 0.05 *, *p* < 0.01 **, and *p* > 0.001 ***.

**Figure 6 pharmaceutics-15-01028-f006:**
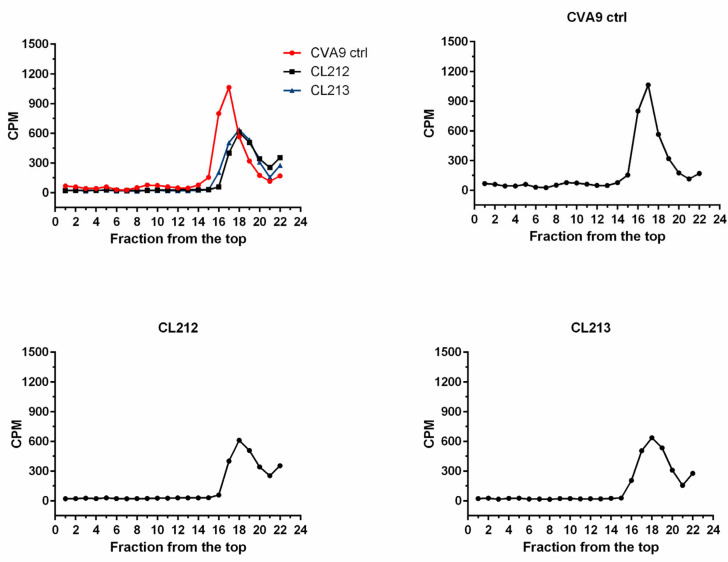
The effect of CL212 and CL213 on the virus capsid using metabolically labeled CVA9. CVA9 was treated with CL212 or CL213 for 1 h at +37 °C and then added on top of a 5–20% sucrose gradient. The fractions were collected from the top and radioactivity was measured as counts per minute (CPM) for each separate fraction.

**Figure 7 pharmaceutics-15-01028-f007:**
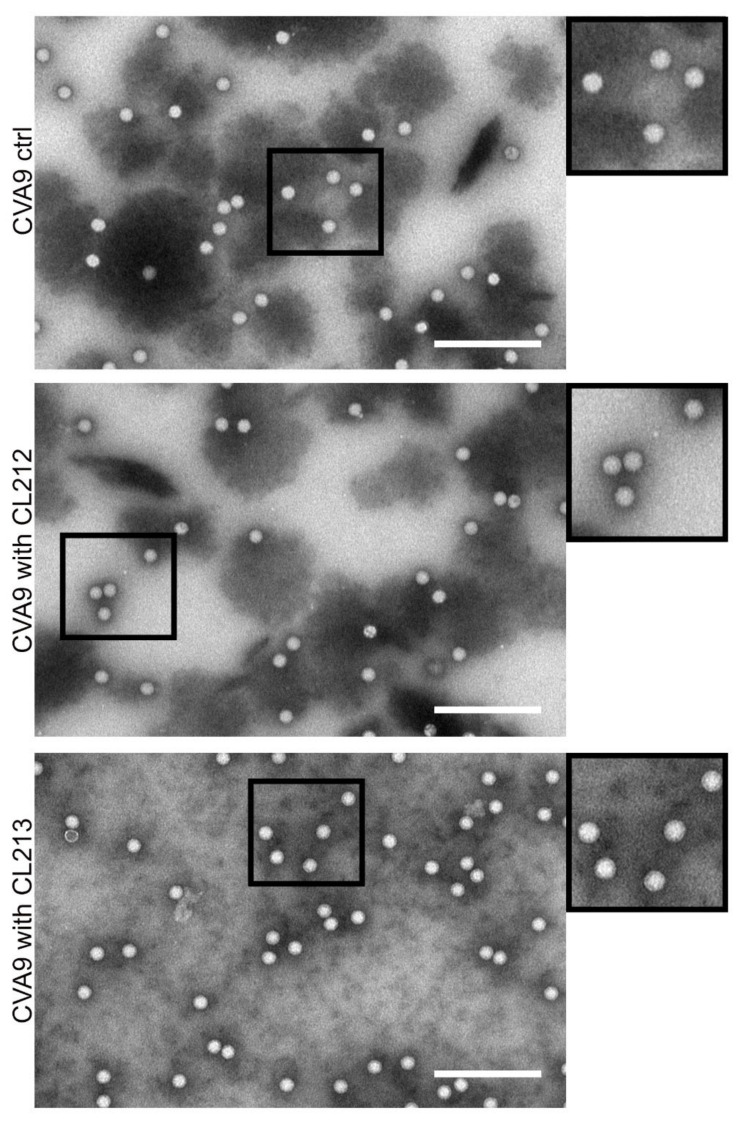
TEM images of CVA9 (1 µg) after being treated with 100 µM CL212 and CL213 for 1 h at +37 °C, and a control with no drug treatment. Scale bars 200 nm.

**Figure 8 pharmaceutics-15-01028-f008:**
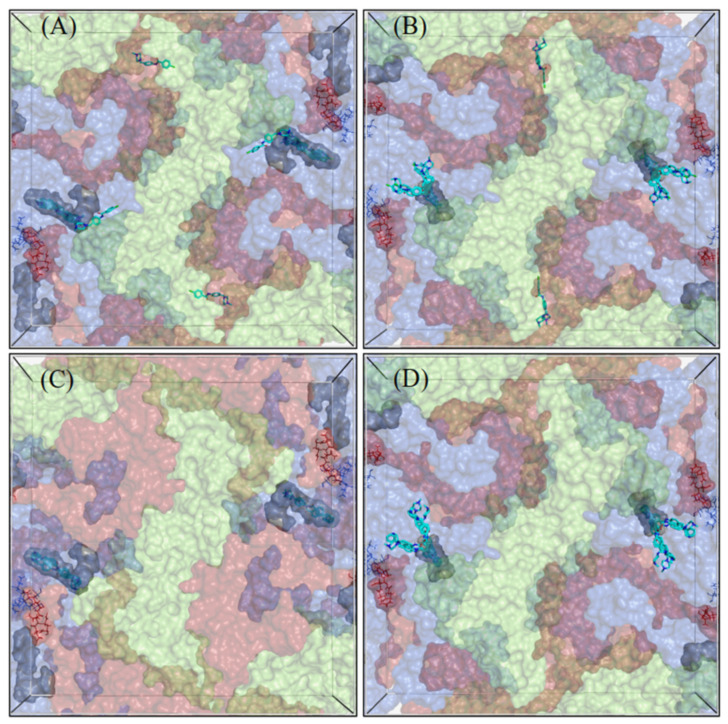
Docking grid boxes centered at the 2-fold axis of the virus capsid. (**A**,**C**) for CVA9 and (**B**,**D**) for CVB3. The docking poses of CL212 (upper panel) and CL213 (lower panel) are represented as cyan sticks. The hydrophobic pocket is represented as a grey surface and the Butcher-Neyts pocket as lines. The capsid proteins VP1–VP3 are colored blue, light green, and red, respectively. VP4 is not shown.

**Figure 9 pharmaceutics-15-01028-f009:**
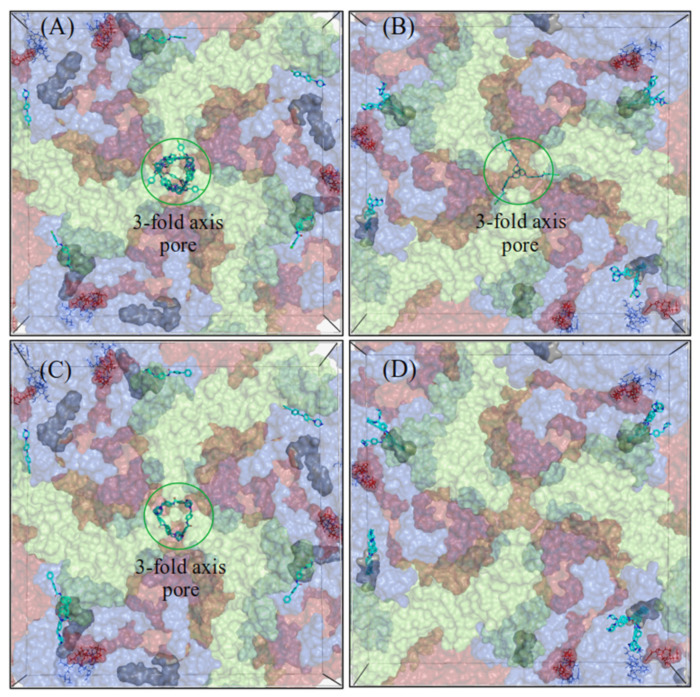
Docking grid boxes centered at the 3-fold axis of the virus capsid. (**A**,**C**) for CVA9 and (**B**,**D**) for CVB3. The docking poses of CL212 (upper panel) and CL213 (lower panel) are represented as cyan sticks. The hydrophobic pocket is represented as a grey surface and the Butcher-Neyts pocket as lines. The 3-fold axis pore is circled in green. The capsid proteins VP1–VP3 are colored blue, light green, and red, respectively. VP4 is not shown.

**Figure 10 pharmaceutics-15-01028-f010:**
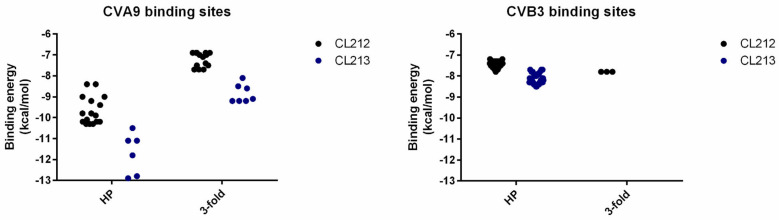
The binding energies of CL212 and CL213 docking poses across CVA9 and CVB3 capsids. The data are presented from two regions: hydrophobic pocket (HP) and 3-fold axis pore (3-fold). Each dot represents one experimental unit, a docking pose.

**Table 1 pharmaceutics-15-01028-t001:** EC_50_ and CC_50_ values of CL212 and CL213 against CVA9 on A549 cells. The selectivity index was calculated by CC_50_/EC_50_.

Molecule	EC_50_	CC_50_	SI
CL212	5.92 µM	>1000 µM ^1^	>168.92
CL213	1.09 µM	152.05 µM	139.50

^1^ Exact value could not be determined due to concentration limitations.

## Data Availability

Not applicable.

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
