# Peer review of "Antiviral Mechanisms of N-Phenyl Benzamides on Coxsackie Virus A9"

_pharmaceutics, 2023, doi:10.3390/pharmaceutics15031028_

Round 1

Reviewer 1 Report

The reviewed publication concerns the study of two new potential antiviral compounds CL212 and CL213. Research was conducted towards two strains of Coxsackie  viruses CVA3 and CVB9. The work is interesting, but it is not entirely clear why the authors chose these two (out of numerous serotypes of this virus) strains of the virus, whether it has any medical justification (high virulence of these strains, specific diseases caused).

Over 200 compounds were analyzed in the publication, however, there is no information about the remaining analyzed compounds (structure and information about activity) in such a case, information about the lack of specific antyviral activity is also the result of research. In my opinion, it would be worth placing the structures of all the analyzed compounds (with antiviral activity) in supplementary materials.

At the same time, the discussion of SAR (3.2) is very limited and it seems that at this point it is worth discussing this analysis in a broader aspect, especially since such activity is usually not binary (active, inactive) but the level of activity varies. Therefore, in this aspect, it would be useful to present the results of antiviral activity for the remaining compounds.

Research using docking methods are not (in my opinion) clearly presented, in the work I could not find a clearly defined (e.g. PDB code) spatial structure of the molecular target, it is not entirely clear where this structure comes from.

Reviewer 2 Report

In this work, the authors showed that CL212 and CL213 efficiently inhibited the infection of CVA9 with low micromolar concentration. They further demonstrated that the compounds acted as capsid binders with suggested localization into the hydrophobic pocket and 3-fold axis area. The work is important for the development of novel antivirals, and the results are interesting. I recommended it to be published in Pharmaceutics after minor revision.

They have screened an in-house chemical library of 200 compounds of diverse chemical scaffolds against CVA9 and CVB3. It is recommended to put the activity data table of all compounds in SI. In addition, the sources of 200 compounds need to be explained. Are they purchased or synthesized by reference?

I want to known whether the compounds interact with RNA?

For “2.8 Docking imaging”: Need to add process description.

The EC50 value is obtained by what software. It needs to be explained. The results given by different software will vary.

References are missing page information.

Round 2

Reviewer 1 Report

The publication has been partially revised - as suggested. The authors wrote "We have now added the chemical structures of all compounds as well as their antiviral activity in the supplementary materials (Table S1, Figure S2)."however, I did not find the attached Figure S2 in the supplementary materials, and in Table S1 there are informations about the SwissADMET analysis, not about antiviral activity.

Author Response

We apologize that the revised supplement material was not properly uploaded in the system. It can be found in the manuscript section inside the ''manuscript.zip'' file which contains both revised manuscript and revised supplement material. For some reason it was not uploaded in the supplement section. We are sorry for the inconvenience.